



# Simulation and Field Campaign Evaluation of an Optical Particle Counter on a Fixed-Wing UAV

Joseph Girdwood[1], Warren Stanley[1], Chris Stopford[1], and David Brus[2]

[1]Centre for Atmospheric and Climate Physics, School of Physics, Astronomy and Mathematics, University of Hertfordshire, Hatfield, Hertfordshire, AL10 9AB
[2]Atmospheric Composition Research, Finnish Meteorological Institute, PO Box 503, FI-00101, Helsinki, Finland

**Correspondence:** Joseph Girdwood (j.girdwood@herts.ac.uk)

**Abstract.** Unmanned aerial vehicles (UAVs) have great potential to be utilised as an airborne platform for measurement of atmospheric particulates and droplets. In particular, the spatio-temporal resolution of UAV measurements could be of use for the characterisation of aerosol, cloud, and radiation (ACR) interactions, which contribute to the largest uncertainty in the radiative forcing of climate change throughout the industrial era (Zelinka et al., 2014). Due to the infancy of the technique however, UAV-instrument combinations must be extensively validated to ensure the data is of high accuracy and reliability. This paper presents an evaluation of a particular UAV-instrument combination: the FMI-Talon fixed-wing UAV and the UCASS open-path optical particle counter. The performance of the UCASS was previously evaluated on a multi-rotor airframe by Girdwood et al. (2020). However, fixed-wing measurements present certain advantages—namely endurance, platform stability, and maximum altitude. Airflow simulations were utilised to define limiting parameters on UAV sampling—that is, an angle of attack limit of 10° and a minimum airspeed of 20 ms$^{-1}$—which were then applied retroactively to field campaign data as rejection criteria. The field campaign involved an inter-comparison with reference instrumentation mounted on a research station, which the UAV flew past through stratus cloud. The effective diameter measured by the UAV largely agreed within 2 $\mu$m. The droplet number concentration agreed within 15% on all but 5 profiles. It was concluded that UCASS would benefit from a mechanical redesign to avoid calibration drifts, and UAV attitude variations during measurement should be kept to a minimum.

## 1 Introduction

Paradoxically, while clouds and aerosols are some of the most studied atmospheric phenomena, their microphysical properties—and overall effect on the earth—are still not completely understood. The Intergovernmental Panel on Climate Change (IPCC)—an important avenue for scientific communication to policy-makers—has, in its fifth assessment report, cited both aerosol radiative forcing and the cloud feedback mechanism as a major uncertainties in climate prediction (IPCC, 2013). These uncertainties are magnified further when we consider that the evolution and optical properties of a cloud depend on the chemistry, morphology, and concentration of the cloud condensation nuclei (CCN)—thus coupling the aerosol and cloud feedback uncer-



tainties. In addition, precipitation initiation involves multiple mechanisms which lack adequate explanation—particularly in warm clouds (Laird et al., 2000; Small and Chuang, 2008; Beard and Ochs, 1993), but also in mixed-phase and ice clouds.

Since cloud microphysical evolution through to precipitation depends strongly on both upwelling and downwelling radiative fluxes, aerosol, cloud, and radiation (ACR) effects are coupled. This would therefore also influence the precipitation initiation through radiation effects discussed recently by Zeng (2018). The contributions to climate change by these ACR interactions are discussed by Zelinka et al. (2014), where it is stated that they contribute the largest uncertainty in the radiative forcing of climate change throughout the industrial era. Zelinka et al. (2014) also states that models which include aerosols in ice

clouds tend to contain large radiative anomalies. Since clouds can be heterogeneous on all length scales down to 1 cm, and time scales in order of seconds (Blyth, 1993), observations with high spatio-temporal resolution need to be performed to characterise ACR interactions. Several previous studies have attempted to characterise the sign and magnitude of the ACR contribution to the overall radiative forcing effect (for example Douglas and L'Ecuyer, 2019). However, these studies rely on remote sensing and modelled data, which results in large uncertainties. A mechanism for high-resolution, in-situ observations

of cloud microphysics can reduce these uncertainties.

Conventionally, in-situ measurements of clouds are conducted using large manned aircraft. These remain the most well characterised and proven platforms for airborne atmospheric measurement, bolstered by large-scale studies into reliability and potential artefacts (Korolev et al., 2013a, b; Spanu et al., 2019). However, large maintenance and running costs for such platforms make the use of manned aircraft only feasible for a limited number of large-scale campaigns. In addition, most

aviation governing bodies do not allow manned aircraft to fly below a certain altitude; the UK Civil Aviation Authority (CAA), for example, define the lower altitude limit as 500 ft (150 m)—making a large amount of atmospheric aerosol, and low level cloud, legislatively impossible to measure this way.

The problem is not only one of measurement quality, but of measurement quantity. Manned aircraft, and remote sensing techniques, are facets in a larger solution, necessitating—in addition to this—a method for the repeatable, accurate sampling of

cloud droplets and aerosol in the lower portion of the atmosphere. The current lack of such a method is one of the reasons for the uncertainty in the radiative forcing of climate change, since this region in particular plays host to many clouds, and much of the total atmospheric aerosol mass.

Unmanned aerial vehicles (UAVs) have gained recent popularity as an atmospheric measurement platform. Generally, these boast distinct advantages over conventional measurement platforms, since they are low-cost, lightweight, and highly

manoeuvrable—providing exceptional spatio-temporal sampling capabilities. However, largely due to the infancy of the UAV as a measurement platform, and the lack of instrumentation specifically designed for UAVs, these require extensive validation in order to ensure observations are accurate and reliable. Girdwood et al. (2020) shows the design and validation of the combination of a multi-rotor UAV and custom-built optical particle counter (OPC): an instrument class commonly used to study cloud and aerosol microphysics. This study used an adaptation of the Universal Cloud and Aerosol Sounding System (UCASS,

Smith et al., 2019). Originally designed for use on meteorological balloons as part of a sounding system, the UCASS utilises a novel naturally aspirated flow transportation mechanism which allows the negation of most aerodynamic artefacts resulting



from UAV airframes, for example anisokinetic flow. The UCASS-UAV combination, therefore, is a good starting point for the characterisation of cloud evolution and ACR interactions.

Fixed-wing UAVs have several advantages over multi-rotors which render them more suitable in certain applications. For
example, fixed-wings have greater flight stability, have a longer endurance and range, can reach higher altitudes, benefit from less serious consequences of icing, and induce a less complex airflow with a lower turbulent kinetic energy. These features make fixed-wings more suited for measurements where larger altitudes are desired, and in locations with access to a runway, or a suitable analogue. Therefore, this study will focus on the evaluation of UCASS on a fixed-wing platform, and assess any resulting artefacts.

## 2 Description of UCASS and UAV

### 2.1 UCASS, Data, and Limitations

Figure 1 is a cross-section view of UCASS, and an illustration of its working principle. As a particle travels through the UCASS unit, it has a chance of passing through the sample area. This area is optically defined by the depth of field (DoF) and field of view (FoV) of the dual-element photodiode detector, which is labeled on Fig. 1. When a particle or droplet passes through
this region, it is illuminated by a laser—highlighted red in Fig. 1—which causes light to be scattered onto an elliptical mirror. The first focal point of the mirror is located at the centre of the sample volume, while the second is located at the centre of the photodiode, thus meaning light scattered by the particle is focused on the detector. The current produced by the detector is proportional to the size of the particle or droplet. Further technical detail of the UCASS can be found in Smith et al. (2019).

A concentration can be derived from UCASS data using one of two methods: particle time of flight (ToF) through the
UCASS laser beam—the width of which is fixed and known, and the velocity of the UCASS itself—normally measured using a pitot tube. The former has the advantage of being the most direct measurement of sample area velocity, while the latter has the advantage of reliability and independence from particle size. The dependence of ToF on particle size originates from the method by which the UCASS electronics calculates it—that is, the time between a pulse rising and falling above and below a threshold amplitude. Since the pulses can be assumed to be roughly Gaussian, two pulses with different maximum amplitudes
through the same laser beam thickness would rise and fall above and below any threshold at different times. This difference is greater with threshold voltage height, and since the threshold must be above normal noise levels the variation is significant with increased particle size. Figure 2 shows an illustration of this effect. The concentration derivation methods used are discussed further in Sect. 3 and Sect. 4.

The data from UCASS are available through its serial peripheral interface (SPI). Each data-point consists of the following
for a given time interval: the particle counts in each of the 16 size bins; the mean particle ToF for bins numbered 1, 3, 5, and 7; the sampling period—the amount of time UCASS records particles for a datum; and some debugging information. The debugging information includes: The amount of particles with an extremely short or long ToF, which is used to filter out noise; the cumulative sum of all the particles counted, which is used to verify the data has been logged properly; and the amount of





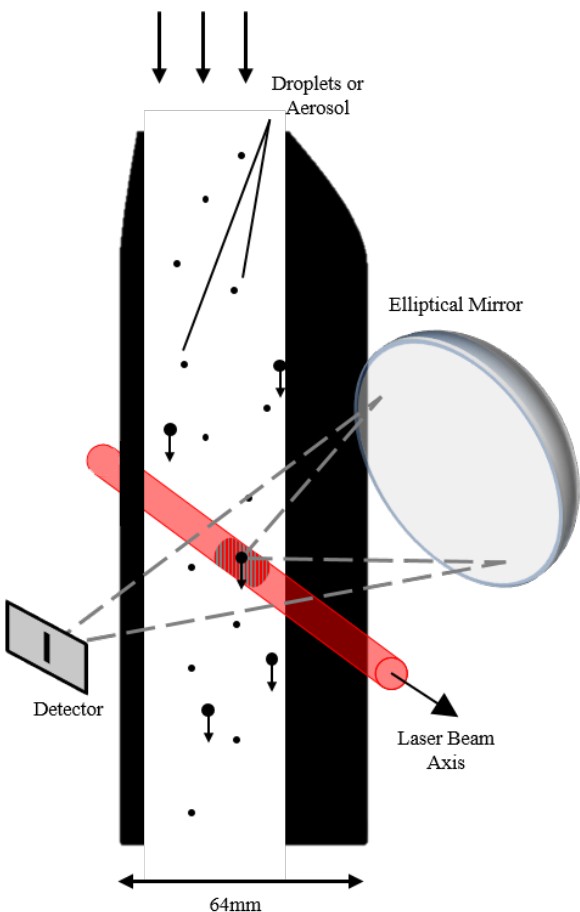

**Figure 1.** An illustration of the UCASS working principle: scattered light at the first focal point of the elliptical mirror is reflected onto its second. The magnitude of this light is proportional to the size of the particle causing the scattering.

particles which have been rejected due to being outside of the sample volume, which is used to determine if co-incidence is
likely.

The three main limitations of the UCASS – fixed-wing UAV combination are airspeed, angle of attack (AoA), and endurance. Airspeed is a limitation due to the bandwidth of the transimpedance amplifier (TIA) which converts the current from the photodiode into a scaled voltage. The cut-off frequency of the TIA is approximately 417kHz, meaning a the maximum pulse width of the signal produced by a particle scattering light onto the photodiode is 2.4 $\mu$s. Since the thickness of the laser beam
at the sample area is 60 $\mu$m, a maximum airspeed of 25 ms$^{-1}$ was allowed in these experiments. Smith et al. (2019) showed that the particle number concentration measured by UCASS deviated from that measured by reference instrumentation when the airspeed increased above 15 ms$^{-1}$. This was likely due to the drop off in amplification by the TIA as the frequency approached



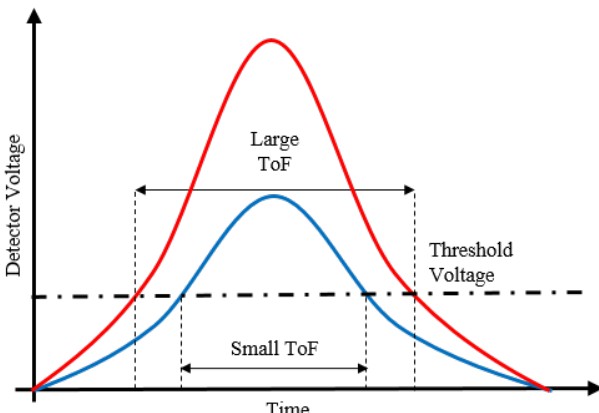

**Figure 2.** An illustration of the ToF dependence on particle size for UCASS measurements. The red and blue lines represent large and small particles respectively for the same laser beam thickness.

the cut-off. For this reason, the ideal airspeed for these experiments was treated as 15 ms$^{-1}$, however this was not always possible due to environmental and engineering factors, which are discussed in later sections.

The AoA—that is, the angle between the UCASS axis and inlet airflow vector—limit for the UCASS unit was defined in Smith et al. (2019) to be ±10°. However, this assessment was only conducted for the UCASS alone, while not mounted on a platform. Therefore, this limit needed to be re-assessed for these experiments, the details of which are discussed in Sect. 3.

Since UCASS was originally designed as a single-use instrument, the endurance must be questioned when operating a UCASS unit consistently throughout a research campaign. Specifically related to fixed-wing UAV operation, stresses from

landings are likely to have the largest impact. Girdwood et al. (2020) introduces a ruggedised re-design of UCASS for multi-rotor UAVs. However, the stresses exerted on the instrument by a multi-rotor design are much higher than those exerted by a fixed-wing. Also, this design could not be directly translated onto a fixed-wing due to its size, since—unlike in the original UCASS—the laser and optics carrier protruded perpendicular to the chassis. For those reasons, the original design was chosen for fixed-wing use, but the endurance throughout the campaign was assessed and is presented in Sect. 4.

**2.2 UAV and Mounting**

The UAV chosen for the evaluation of the nose-mounted configuration was the Talon UAV (X-UAV, Tian-Jie-Li Model Co., Ltd., China). The Talon platform was equipped for atmospheric measurements by the Finnish Meteorological Institute (FMI), with consultation from the University of Hertfordshire (UH) regarding the mounting of UCASS. This platform was chosen primarily for its flat-top and wide fuselage, which provided an adequate mounting surface for the UCASS. The wide fuselage

also provided enough space to mount the avionics, which included: a Pixhawk 1.2 flight controller (3DRobotics, 2013) with a global positioning system (GPS) and magnetometer; a Raspberry Pi Zero (RaspberryPi-Foundation, 2015) for the logging





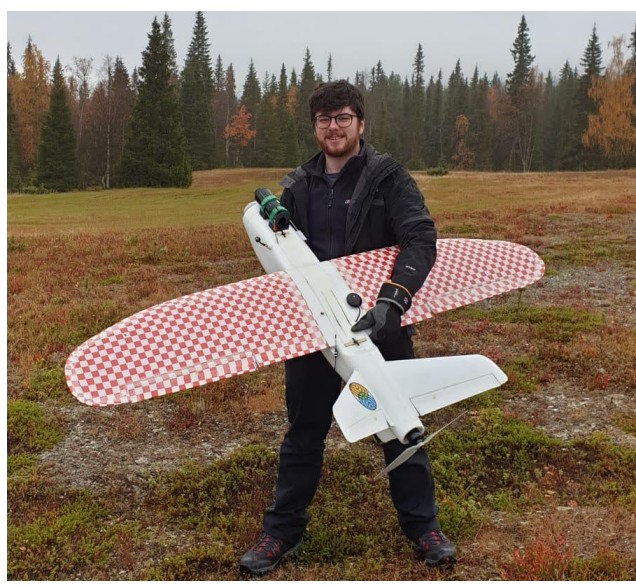

**Figure 3.** The Finnish Meteorological Institute (FMI) 'Talon' UAV, which was used to test the performance of UCASS on a fixed-wing UAV.

of UCASS data; a BME280 temperature, pressure, and humidity sensor; and a pitot-static tube to record the airspeed data essential for UCASS concentration measurements. The temperature and humidity measurements are not evaluated in this paper since they do not relate to the performance of UCASS, but are mentioned for the purpose of transparency, since they

are often accompanying variables in datasets. The capabilities of the FMI-Talon are summarised in Table 1. The UCASS unit was mounted on the nose cone, with its inlet protruding well out of the fuselage boundary layer. This was influenced by the CFD-LPT experiments presented in Sect. 3.

In order to eliminate airframe-induced aerodynamic artefacts, the UCASS unit was positioned on mounts above the nose-cone. The UAV brushless electric motor was rear-facing and positioned at the back of the aircraft behind the tail, therefore no

aerodynamic artefacts resulting from the propeller wash were expected. In addition to this, the position of the motor means that

**Table 1.** Technical specification for the FMI-Talon UAV

| Airframe | Talon |
|---|---|
| Flight Controller | Pixhawk 2.1 |
| Wing Area | 0.6 m$^2$ |
| Wingspan | 1718 mm |
| Length | 1100 mm |
| Maximum Take-off Mass | 3 kg |
| Aircraft Mass Used Here | 2.7 kg |





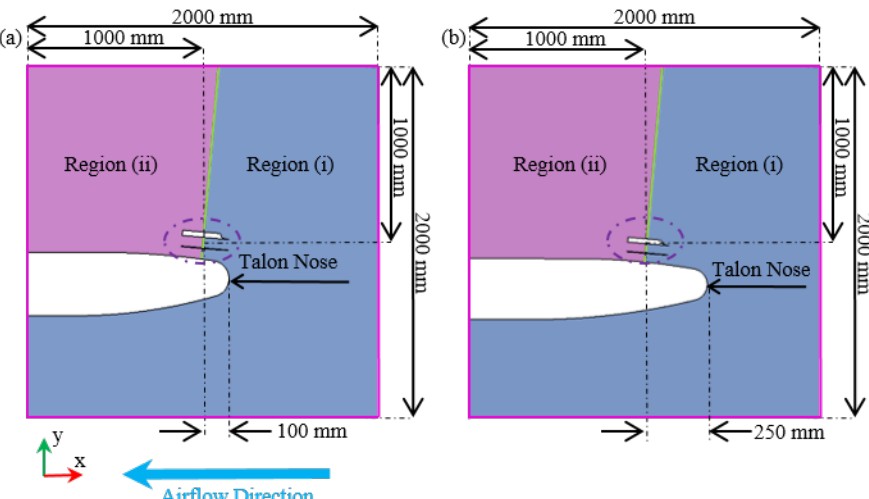

**Figure 4.** Illustrations of the CFD-LPT domains for the two investigated configurations. Figure 4a shows the fore configuration and Fig. 4b the aft. The airflow direction and geometrical axes shown in 4a are consistent throughout. The UCASS is shown in the purple dashed circle, and the FMI-Talon nose cone is indicated. The purple lines represent free-stream boundaries, and the green line represents the divide between regions (i) and (ii)—which is where the LPT droplets are sampled.

no artefacts due to the charging—and subsequent deflection—of aerosols by its oscillating magnetic field are expected. The expected influence of the airframe aerodynamics on measurements, and the exact positioning of UCASS, is discussed in Sect. 3.

## 3 Computational Fluid Dynamics with Lagrangian Particle Tracking

### 3.1 CFD-LPT Method

The presented simulations were conducted using the Star-CCM+ commercial code. Two main mounting configurations were considered for the simulations: a configuration whereby the UCASS was positioned as far forward as the centre of gravity (CoG) tolerance of the FMI-Talon allowed, and a configuration were the UCASS was positioned further back from the end of the nose cone. Hereafter, these will be referred to as the fore and aft configurations respectively. The former was expected to be advantageous only if the UCASS could be positioned far enough forwards to escape aerodynamic influence from the nose cone. The latter was expected to suffer from aerodynamic artefacts resulting from a non-zero AoA—that is, the angle between the axis of UCASS and the airflow velocity vector.

The domains for the fore and aft configurations are shown in Fig. 4a and Fig. 4b respectively. The fore configuration shows the UCASS positioned as far forward as the CoG tolerance allowed. The domain was a two-dimensional slice through the





FMI-Talon, bisecting the centre of UCASS and the UAV nose—indicated with the purple dashed circle and the black arrow in Fig. 4 respectively. The domain was a 2 m by 2 m square, which was found by trial and improvement to suffer from no artefacts due to pressure wave reflections at the desired airspeed. The main point of interest for the simulations was the UCASS sample area, which was made the centre-point of both domains. In both domains, the air flowed in the direction of the blue arrow shown in Fig. 4a, and LPT droplets were injected from the right hand wall. Since droplets are sampled when they cross a

boundary, a boundary line was set up in the domain between regions (i) and (ii)—marked with the green line in Fig. 4—which crossed the UCASS sampled volume. The pink lines around the perimeter of both domains represent 'free-stream' regions which are treated as infinitely extending regions with constant temperature and pressure—defined here as 280 K and 980 hPa respectively.

The same mesh input parameters were used for both fore and aft configurations, since the differences in simulation con-

vergence, simulation time, and accuracy were found to be to be small for the same mesh. A similar mesh dependency test to that presented in Girdwood et al. (2020) was conducted, but the full results are not shown here for the sake of brevity. The chosen base cell size of 10 mm, refined to 1mm in the region surrounding UCASS and by the walls, was influenced by the mesh dependency test results. A polyhedral mesh was chosen since initial tests showed better convergence when compared to tetrahedral or hexahedral.

Since no motion needed to be simulated, the simulation was time-steady—meaning it was solved for one instance and iterated until converged. The simulation was considered to be converged when the continuity residual—that is, the difference in the average imbalance in continuity between adjacent solution iterations—was less than $1 \times 10^{-4}$. Since direct numerical simulation was deemed to be too computationally expensive, a Reynolds-averaged Navier-Stokes (RANS) continuum model with a turbulence parameterisation model was used. The turbulence parameterisation model used was $k-\omega$, which was chosen

due to superior estimates of near-wall turbulence when compared to $k-\epsilon$. The simulation and mesh parameters are summarised in Table 2.

**Table 2.** Parameters used for the CFD-LPT simulations

| | |
|---|---|
| Software | Star CCM+ |
| Continuum Model | RANS |
| Turbulence Model | $k-\omega$ |
| Solver | Time steady |
| Wall Treatment | low-y+ |
| Mesh Type | Polyhedral |
| Max. Mesh Aspect Ratio | 1.2 |
| Maximum Mesh Size | 10 $\mu m$ |
| Droplet Material | Water |



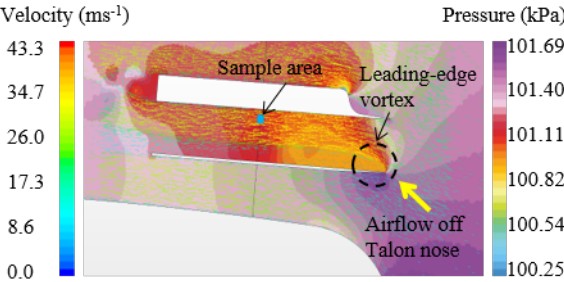

**Figure 5.** A plot of the pressure and velocity fields of the fore simulation, with an AoA of $0°$. The colour of the background is the absolute pressure of the continuum, and the colour and direction of the glyphs show the velocity field. The black circle indicates a leading-edge vortex at the UCASS inlet, the blue square indicates the UCASS sample volume, and the yellow arrow shows the airflow direction at the UCASS inlet.

## 3.2    CFD-LPT Results and Discussion

Since the aft configuration—shown in Fig. 4b—was only intended to be used if the UCASS could not be positioned out of the nose-cone influence due to CoG limits, the fore simulation was conducted first with an angle of attack of $0°$. Figure 5 shows a
magnification of the pressure and velocity fields around UCASS and the FMI-Talon nose cone. The colour of the background is the absolute pressure of the continuum, and the colour and direction of the glyphs show the velocity field. The airflow deflection off the nose cone of UCASS, as indicated by the yellow arrow in Fig. 5, was sufficiently large to create a leading edge vortex on the lower half of the UCASS inlet, which is indicated by the black dashed line. As shown in the background pressure plot of Fig. 5, this leading edge vortex compressed the airflow at the sample volume—indicated by the blue square—which caused
the velocity to increase to 31 ms$^{-1}$. As previously stated in Sect. 2.1, this velocity is well beyond what UCASS is able to tolerate due to the TIA bandwidth, therefore the fore configuration was abandoned at this stage. Droplets were not simulated for this geometry due to the inability of UCASS to measure them even if their trajectory and theoretical concentration would be unaffected by the leading edge vortex. This non-ideal positioning could have been subverted by the use of a custom built airframe—similar to Mamali et al. (2018); Girdwood et al. (2020)—which would have more flexibility around positioning and
could be designed around UCASS. However, the design and construction of a custom airframe would be intensive and was beyond the scope of this research.

The remaining simulations were conducted using the aft geometry in Fig. 4b. Since it was predicted that the aft configuration would be more sensitive to AoA changes, the first experiment was designed to keep droplet size and continuum velocity constant, while varying the free-stream velocity angle with respect to the FMI-Talon nose. The droplet diameter used was 20
$\mu$m, which was chosen to be slightly above a typical modal value of the droplet size distribution in a mature stratus cloud. Since the AoA limit for the UCASS without a platform is $10°$, the AoA range chosen was $-20°$ to $20°$—since an AoA of $\pm20°$ was deemed unlikely to be acceptable. Using Fig. 4 as a reference, a negative AoA was considered to be one which



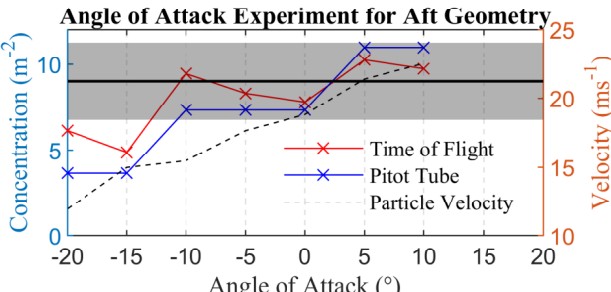

**Figure 6.** Calculated droplet concentration against AoA for CFD-LPT simulated data. The solid black line shows the input concentration, and the black shaded region around it represents $\pm25\%$ of the input. The red and blue lines are the concentration data calculated using simulation equivalent droplet ToF and pitot tube airspeed respectively. The black dotted line is the mean velocity of a droplet as its simulated trajectory crossed the UCASS sample area.

points downwards towards negative '$y$'. An AoA step of $5°$ was chosen as a balance of computation time and resolution. The injected concentration of droplets was 9 m$^{-2}$, which was determined by experimentation to be the lowest concentration that

could provide a statistically significant number of droplet streams at the UCASS sample volume. The value of 9 m$^{-2}$ was largely abstract since the simulations were time steady and therefore conducted for one specific instance. The droplets were therefore simulated as parcel streams with inertia. The injected concentration of 9 m$^{-2}$ corresponded to 10 parcel streams per 100 mm at the simulation inlet. The airspeed used for these simulations was 20 ms$^{-1}$ since this was the target maximum airspeed for the UAV.

Since not enough droplets could be simulated to for a statistically significant quantity at the true 0.5 mm sample area of UCASS, a droplet was considered to have crossed the UCASS sample area if its trajectory crossed a plane coincident with the sample area, within 10 mm of its origin. As discussed previously in Sect. 2.1, there are two possible definitions of concentration from real UCASS data: ToF and pitot tube. Both definitions had to be assessed for differences, and therefore re-defined in the context of these CFD-LPT simulations. Droplet number concentration is simply defined here as

$$N_c = \frac{n_{s10}}{A \times v \times t} \tag{1}$$

where $N_c$ is the simulated number concentration (m$^{-2}$), $n_{s10}$ is the number of droplet parcel streams which intersect the sample plane within 10 mm of the sample area origin, $v$ is a velocity (ms$^{-1}$), and $t$ is time (s)—which is an arbitrary scaling factor in this instance since time is steady. For all simulations, $t$ was set equal to 1 for both the input and calculated output concentrations. The velocity ,$v$, used depended on the sample volume calculation method. For the ToF equivalent, the droplet velocity at the

sample plane was used; for the pitot tube, the freestream velocity of 20 ms$^{-1}$ was used.

Figure 6 shows a plot of concentration and droplet airspeed through the sample volume versus AoA for the simulation. The red and blue lines are the concentration versus AoA calculated using the ToF and pitot tube methods respectively, and the black dotted line is the mean velocity of the droplets as they cross the sample plane. The solid black line is the injected concentration



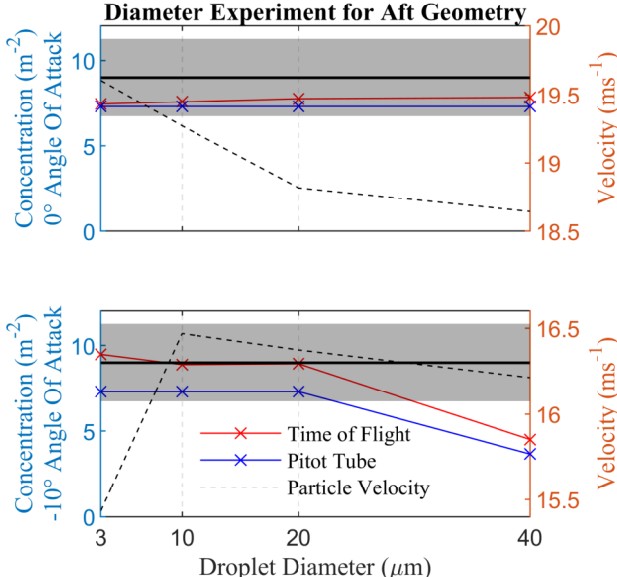

**Figure 7.** Calculated droplet concentration against droplet diameter for two different AoAs. The solid black line shows the input concentration, and the black shaded region around it represents ±25% of the input. The red and blue lines are the concentration data calculated using simulation equivalent droplet ToF and pitot tube airspeed respectively. The black dotted line is the mean velocity of a droplet as its simulated trajectory crossed the UCASS sample area.

of droplets at the inlet, and the shaded region is ±25% of the inlet concentration which was considered an acceptable error

margin for concentration. This was because a light-weight OPC agreeing with a reference within this margin is generally considered robust, for example in Clarke et al. (2002). It was observed in the simulations that the concentration of droplets generally increased with AoA until 10°, after which no droplet trajectories at all reached the sample area. This was because any AoA above this would cause boundary layer separation from the upper part of the fuselage, thus putting it under stall conditions and causing the majority of the droplets to be deflected around the resulting vortex.

The general increase of droplet concentration with AoA is due to the convergence of continuum streamlines towards the upper part of UCASS before the boundary layer separates. This effect both increased the convergence of droplet parcel streams— and thus the pitot tube estimated concentration—and the particle velocity at the sample plane. These effects conflict in their influence on the ToF particle concentration, since as velocity increases, the concentration calculated from Eq. 1 will decrease. This caused an increase in accuracy of the ToF concentration, which was within ±15% of the injected concentration for AoAs

between -10° and 10°.

  Both ToF and pitot tube derived concentrations are found to be within acceptable limits, within the AoA range of -10° and 10°. This is the same range as the UCASS without factoring in platform artefacts (Smith et al., 2019).





In addition to the AoA dependency, droplet size dependency was also tested in order to ensure the shape of the size distribution was not affected. Since it was predicted that increased AoA would amplify size dependency, the simulations were run for

AoAs of 0° and -10°. The latter limit was chosen since, for negative AoAs, the inertia of larger droplets was predicted to force them away from the sample volume, which was closer to the upper edge of UCASS. The sizes used for the simulation were 3, 10, 20, and 40 $\mu$m, which were chosen since this range spanned the full size range of the UCASS in low-gain mode, and was distributed approximately logarithmically. The results of this simulation are shown in Fig. 7, where the top panel shows the results for 0°, and the bottom shows the results for -10°. The graph legend is consistent with Fig. 6. The results of this

simulation show that, for an AoA of 0°, the size dependency is minimal—this can be seen in the top panel of Fig. 7. During these simulations—for an AoA of 0°—the particle velocity did not deviate much from 20 ms$^{-1}$, meaning the ToF and pitot tube derived concentrations were found to be similar, with a slight deviation for a size of 40 $\mu$m due to the slower acceleration within the UCASS inlet.

The size dependency results for an AoA of -10° are shown in the bottom panel of Fig. 7. This experiment revealed a larger

concentration discrepancy for 40 $\mu$m droplets, which resulted from their larger inertia deflecting them away from the sample area. This discrepancy was beyond the $\pm 25\%$ limit which was considered acceptable. Beyond a re-design of the UCASS aerodynamics, however, there was little methodology which could rectify this. A large uncertainty of $\pm 45\%$—the percentage difference between the measured and input concentration of 40 $\mu$m droplets—is therefore placed on measurements of droplets above 30 $\mu$m, which is where the ToF derived concentration strays beyond limits using linear interpolation.

The CFD-LPT simulations showed that ToF was, theoretically, the superior method for determining airspeed, since concentration derived using this method was always closer to the input. However, in practice this was not always the reality. This was because of the dependence of ToF determined airspeed on a fixed beam width, which would change if the UCASS were to be subjected to large mechanical forces that could misalign the optics. In addition—as previously mentioned in Sect. 2.1—ToF measured by the UCASS was dependent on the measured droplet size, so a calibration for each bin had to be applied. Also,

if enough droplet measurements were not performed in a time-step integrated histogram, a statistically significant ToF measurement of that histogram cannot be performed. Pitot tube airspeed was therefore used when there were too few particles to perform a ToF measurement.

## 4 Field Campaign Testing and Validation

### 4.1 Field Campaign Method

The validation flights were undertaken during the Pallas Cloud Experiment (PaCE) in September 2020, at the Pallas atmosphere-ecosystem super-site. This is located 170 km north of the Arctic Circle (67.973°N, 24.116°E), partly in the area of Pallas-Yllästunturi National Park (Lohila et al., 2015). Measurements were taken from the 24[th] to the 30[th] of September, however stratus cloud was only present from the 28[th] to the 30[th]. A fault in the data-logger used to gather flight data was persistent on the 30[th], therefore only flights conducted on the 28[th] and 29[th] will be considered in this analysis. On each day, 4 profiles were

conducted. Each profile consisted of an ascent and a descent through the entire cloud, which was approximately 4 km thick





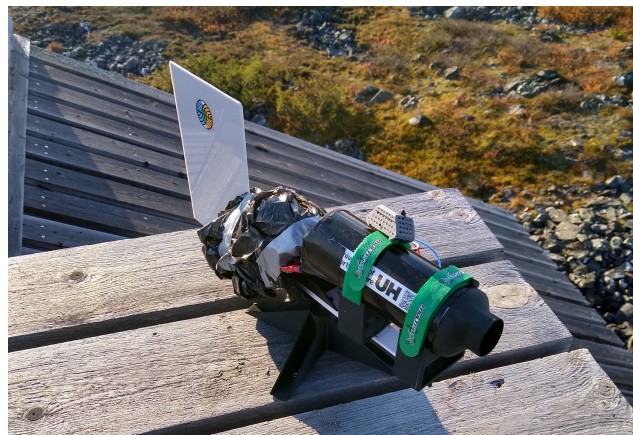

**Figure 8.** An image of the UCASS mounted on the roof of Sammaltunturi station. The instrument was connected to a wind vane and data logger, and re-mounted every morning before the days flights.

throughout the experiments. On the 28[th] of September the mean wind speed throughout the day of flights was approximately 7 ms[-1], the wind direction 200° from north, and the temperature 5°C. On the 29[th] of September the mean wind speed throughout the day of flights was approximately 5 ms[-1], the wind direction 180° from north, and the temperature 8°C.

The layered stratus cloud common to Pallas-Yllästunturi, combined with the airspace D527-Pallas reserved by FMI made
this location perfect for the validation of UAV based optical particle instruments. Since stratus cloud is normally laterally homogeneous within the distance scales discussed in this paper, co-location errors between the UAV and station could be assumed to be minimal. The measured particles within the stratus cloud were all assumed to be water droplets, therefore the calibration lookup table for UCASS was made using a refractive index of 1.31+0i—the refractive index of water. Cloud droplets were considered the ideal validation particle since the UCASS calibration relies on Mie scattering, and the droplets
can be assumed to be spherical. Water droplets also exhibit low absorption of 650 nm light—the wavelength of the UCASS laser. This is beneficial since absorption can cause artefacts in particle size measurements which utilise scattering like UCASS.

For comparative data, instruments were mounted on the roof of Sammaltunturi station, which was located on a hill 565 m above sea level. The instrument chosen for the comparison was an identical UCASS unit, with the same TIA gain, oriented into the wind via the means of a wind vane and turret system, an image of which is shown in Fig. 8. The only differences in the data
between the two UCASS units would be due to platform artefacts, since they were otherwise identical. The aspiration airflow for UCASS was provided by the wind. The minimum airflow required for UCASS operation is 3 ms[-1], since particles which produce longer ToFs than this are rejected in a firmware noise filter. At all instances during the campaign, the wind speed was faster than 3 ms[-1]. The wind speed for each UCASS data-point was measured using an anemometer on Sammaltunturi station. The UCASS – wind-vane set-up was secured to the roof each morning, before the daily flights started, and removed after flying
was stopped for the day. This was to prevent damage to UCASS during the night, which may cause optical misalignment.





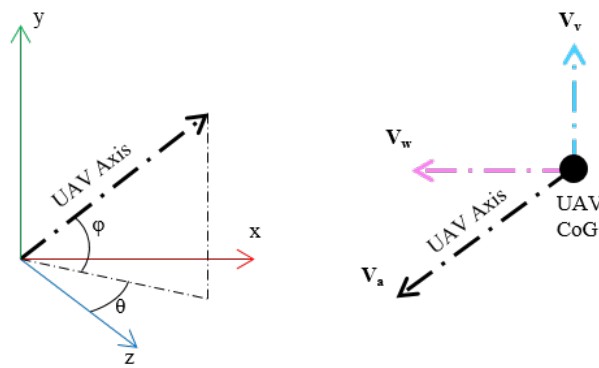

**Figure 9.** An illustration clarifying the angle and velocity variable definitions used for the computation of AoA.

The FMI-Talon ascents and descents were conducted utilising a mixture of spiral patterns and wind-parallel patterns—both of which are commonly conducted in literature (Martin et al., 2011; Mamali et al., 2018). The conducted flight patterns are summarised in Table 3. The purpose of this was to compare the two patterns to determine which is superior for UCASS measurements. Beyond the limits of the stratus cloud, when insufficient particles were present, the pitot tube airspeed was used to derive a sample volume. Within the cloud ToF was used. It was also observed that within the cloud the pitot tube would become blocked, which caused the airspeed reported by the pitot tube to increase beyond physically possible limits. Not only did this render pitot tube measurements of sample volume impossible, the FMI-Talon also had to be manually piloted through the cloud since the large over-prediction in airspeed would cause the autopilot to stall the aircraft. Data from UCASS were recorded at 0.5 s intervals by a Raspberry Pi Zero (RaspberryPi-Foundation, 2015); pitot airspeed, attitude, and GPS data were recorded by the flight controller; and temperature, pressure and humidity were recorded by a separate Raspberry Pi. The flights considered for analysis were the only flights for which all three data loggers were operating.

**Table 3.** A table detailing the profile types of the UAV flights considered for analysis.

| Date | Time | Ascent | Descent |
|------|------|--------|---------|
| 2020/09/28 | 10:42 | Lines | Lines |
| 2020/09/28 | 11:26 | Lines | Lines |
| 2020/09/28 | 12:09 | Spiral | Spiral |
| 2020/09/28 | 12:45 | Lines | Spiral |
| 2020/09/29 | 07:51 | Lines | Spiral |
| 2020/09/29 | 08:21 | Lines | Spiral |
| 2020/09/29 | 08:21 | Lines | Spiral |
| 2020/09/29 | 12:52 | Lines | Spiral |





The data were filtered based on two criteria, the first of which was a maximum airspeed limit, which was set to 20 ms⁻¹. This was set because, even though the cut-off frequency of the TIA corresponds to an airspeed of 25 ms⁻¹, the CFD-LPT simulations revealed that a droplet could travel faster through the UCASS sample area than the free-stream depending on AoA. An airspeed limit of 20 ms⁻¹ ensures that, for all valid AoAs, the airspeed through the sample area never exceeds 25 ms⁻¹. The second was an AoA limit defined as $\pm 10°$, resulting from the CFD-LPT simulations in Sect. 3. AoA was defined here as the angle between the UAV axis, and the resultant airspeed vector. The resultant airspeed vector is equal to the sum of the wind-speed vector, the vertical velocity of the UAV, and the UAV airspeed vector. The UAV airspeed vector was defined as

$$\boldsymbol{V_a} = V_{pit} \left( \frac{\begin{vmatrix} \sin\theta\cos\phi & \cos\theta\cos\phi & \sin\phi \end{vmatrix}}{\sqrt{(\sin\theta\cos\phi)^2 + (\cos\theta\cos\phi)^2 + (\sin\phi)^2}} \right) \tag{2}$$

where $\boldsymbol{V_a}$ is the UAV airspeed vector, $V_{pit}$ is the pitot tube airspeed, $\phi$ is the UAV elevation angle, and $\theta$ is azimuth angle of the UAV with respect to north—these were calculated from the pitch, roll, and yaw angles of the UAV. The angles and velocities are illustrated in Fig. 9 for clarity. Equation 2 is a normalised vector parallel to the UAV axis multiplied by the UAV airspeed ($V_{pit}$). Since wind-speed was considered likely to change as the UAV ascended, the wind angle measured by Sammaltunturi station was assumed constant with altitude, and the difference between the UAV ground-speed and 'xz' component of airspeed was used to calculate the wind-speed vector. This was given by

$$\boldsymbol{V_w} = \frac{\boldsymbol{V_{xz,w}} - \boldsymbol{V_{xz,g}}}{\pi - \cos\alpha_w} \tag{3}$$

where

$$\boldsymbol{V_{xz,g}} = V_g \left( \frac{\begin{vmatrix} \sin\theta & \cos\phi & 0 \end{vmatrix}}{\sqrt{(\sin\theta)^2 + (\cos\phi)^2}} \right) \tag{4}$$

$$\boldsymbol{V_{xz,w}} = \boldsymbol{V_a} \cos\phi \begin{vmatrix} 1 \\ 1 \\ 0 \end{vmatrix} \tag{5}$$

$\alpha_w$ is the wind angle clockwise from north measured by Sammaltunturi station; $\boldsymbol{V_w}$ is the wind-speed vector; $\boldsymbol{V_{xz,a}}$ is the 'xz' component of UAV airspeed; and $\boldsymbol{V_{xz,g}}$ is the UAV ground speed ($V_g$) expressed in vector form. The vertical velocity of the UAV was given by

$$\boldsymbol{V_v} = \begin{vmatrix} 0 & 0 & V_v \end{vmatrix} \tag{6}$$

where $V_v$ is the vertical speed reported by the flight controller. The resultant airspeed vector was then given by

$$\boldsymbol{V_{r,a}} = \boldsymbol{V_v} + \boldsymbol{V_w} + \boldsymbol{V_a} \tag{7}$$

therefore AoA was finally defined as

$$AoA = \arccos\left( \frac{\boldsymbol{V_{r,a}} \cdot \boldsymbol{V_a}}{|\boldsymbol{V_{r,a}}||\boldsymbol{V_a}|} \right) \tag{8}$$





The $V_a$ used in Eq. 8 could be replaced with any vector parallel to UCASS, but this was used since it was already calculated in Eq. 2. The wind measurements produced via this method were not considered accurate nor reliable enough to be utilised in

scientific datasets, simply because they were not validated here, and were likely to suffer from measurement artefacts. However, they were considered robust enough to be used as rejection criteria for the UCASS particle spectrum data.

## 4.2 Field Campaign Results and Discussion

Figures 10 and 11 show a comparison of dn/dlog($D_p$) against droplet diameter, measured by the FMI-Talon and Static UCASS when coincident, for September the 28[th] and 29[th] respectively. The blue and orange bars represent the dn/dlog($D_p$) measured

in each bin of the FMI-Talon UCASS and the Static UCASS respectively. The effective diameters recorded by both UCASS units are also shown on the plot. Effective diameter, as in Korolev et al. (1999), was defined as the ratio of the third and second statistical moments; that is, for binned UCASS data,

$$D_{eff} = \frac{\sum_{i=1}^{16} n_i D_i^3}{\sum_{i=1}^{16} n_i D_i^2} \tag{9}$$

where $D_{eff}$ is the effective diameter, and $n_i$ is the number concentration in bin $i$ which has the geometric mean diameter

$D_i$. The take-off time and date of each flight is also shown in the bottom left corner. In each figure, the left hand column of sub-figures shows data gathered on the ascending profile, and the right hand column shows that gathered on the descending profile. The dn/dlog($D_p$) was calculated from

$$dn/dlog(D_p) = \frac{c_{n,i}}{\log D_{l,i} - \log D_{l,i+1}} \tag{10}$$

where $c_{n,i}$ is the number concentration in bin $i$, $D_{l,i}$ is the lower size bin boundary of bin $i$, and $D_{l,i+1}$ is the lower size bin

boundary of bin $i+1$. The FMI-Talon data used in Fig. 10 and Fig. 11 was averaged over a 40 m vertical distance—centred on the station altitude. The static UCASS data was averaged over 10 s—centred on the instance the Talon passed the altitude of Sammaltunturi station.

### 4.2.1 Measured Diameter

The effective diameter measured by the UAV on the 28[th] of September overall was within 15% of that measured by the static

UCASS. However there was an observed increase in the difference between the UAV and reference data with time, which would normally point to a calibration discrepancy caused by an offset in optical alignment. However, on the 28[th], the effective diameter measured by the static UCASS was smaller. This was unusual because the FMI-Talon UCASS had undergone much larger stresses, which would cause it to measure lower than the static UCASS if a calibration offset in alignment was the case— since it is unlikely that an offset in alignment would *increase* the intensity of light incident to the photodiode. This trend can

be seen in Fig. 12 in the points under the blue line. Laser mode hopping resulting from a temperature change was considered as a possible cause, however the temperature only changed by 0.9°C throughout the day of flights, which was not enough to cause a significant perturbation in laser wavelength, and therefore the measured effective diameter.





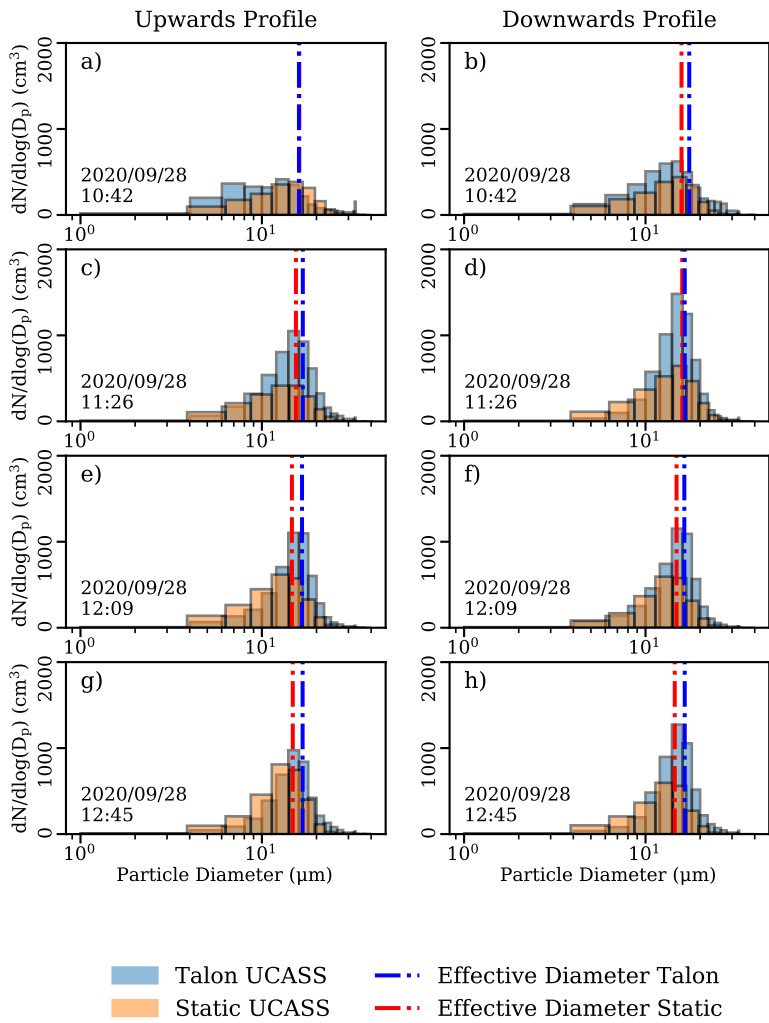

**Figure 10.** Plots of dn/dlog($D_p$) plotted against droplet diameter for the FMI-Talon – UCASS combination coincident with Sammaltunturi station, for the flights conducted on the 28[th] of September 2020. The left hand column of plots is the data taken on the ascending profile and the right hand column is that taken on the descending profile. The thickness of each bar is a UCASS bin, the orange bars are the static UCASS data, and the blue bars are the FMI-Talon – UCASS data. The red dashed line is the effective diameter for the Static UCASS data, and the blue dashed line is that of the FMI-Talon – UCASS data.



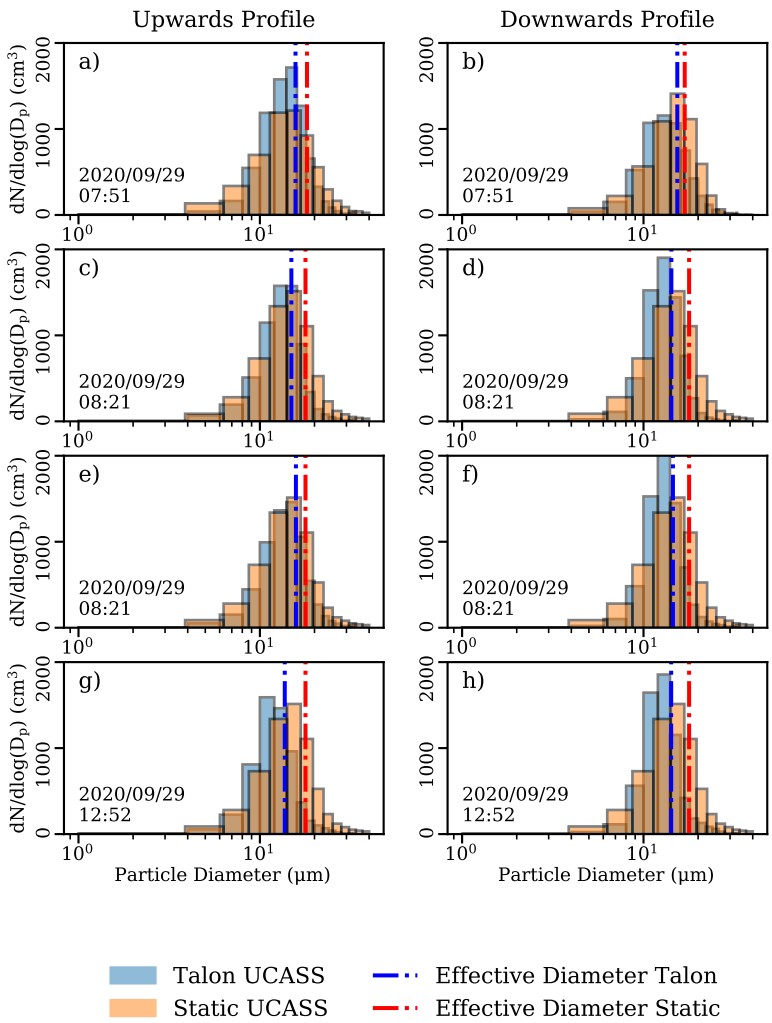

**Figure 11.** Plots of dn/dlog($D_p$) plotted against droplet diameter for the FMI-Talon – UCASS combination coincident with Sammaltunturi station, for the flights conducted on the 29th of September 2020. The legend and axes are consistent with Fig. 10.

Since the static UCASS was left immersed in the stratus cloud for an extended period of time, it was observed that some water droplets accumulated on the elliptical mirror—which was the largest exposed optical surface. These water droplets likely caused extinction of the scattered light from the droplets before it reached the photodiode. The decreased intensity of the photodiode incident light would cause under-sizing of the droplets, which would lead to a gradual decrease in effective diameter



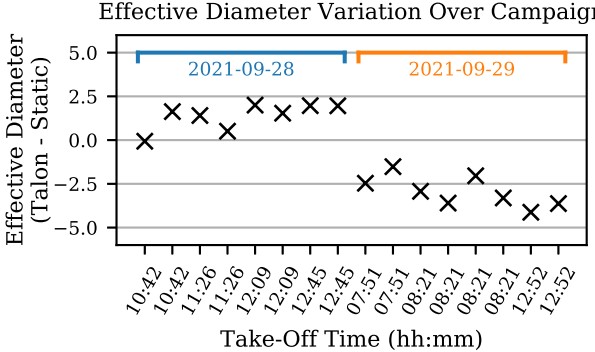

**Figure 12.** A plot of the difference in effective diameter between the station mounted UCASS and the UAV mounted UCASS over the progression of the campaign. The two days discussed are highlighted in blue and orange.

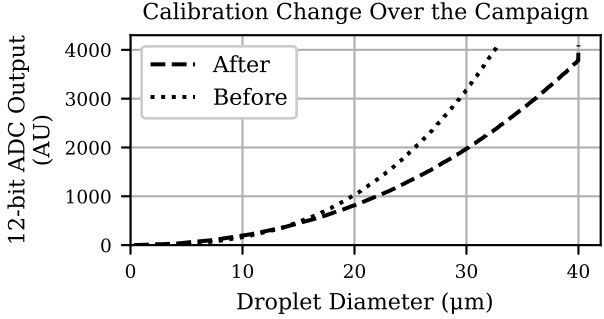

**Figure 13.** A plot of the two instrument response curves—that is, 12-bit ADC output versus scattering cross section diameter—from calibration data collected before and after the campaign.

over time. Since this trend was observed in the data in Fig. 12, this was treated as the most probable cause for the increase of discrepancy in effective diameter over time. When the wind-vane – UCASS rig was re-attached on the station roof on the 29[th], the UCASS was oriented in its mount with the mirror on top to reduce the number of droplets which were deposited on the

mirror via gravitational settling. However, some droplets were still observed to be deposited on the mirror, which indicated that a proportion of the droplets settle via turbulent deposition or droplet shedding. This would require a re-design of the internal UCASS aerodynamics, similar to Korolev et al. (2013a).

On the 28[th] and 29[th] of September, a larger difference in effective diameter between the static UCASS and FMI-Talon UCASS was measured, which can be seen in Fig. 11. Contrary to measurements taken on the 28[th], the static UCASS measured

a larger effective diameter than the FMI-Talon UCASS, which can be more clearly seen in Fig. 12 under the orange line. The most likely cause for this was an offset in alignment in the FMI-Talon UCASS causing less light to be focused on the

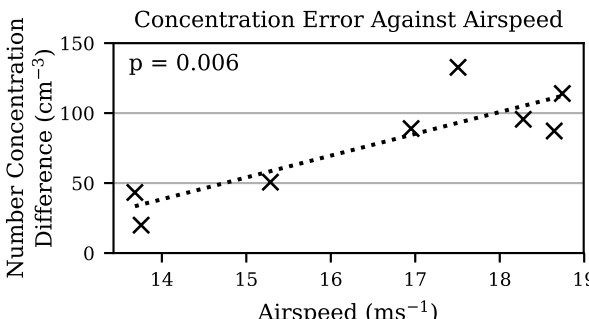

**Figure 14.** A correlation plot of the absolute difference in measured number concentration between the FMI-Talon and Static UCASS units, and airspeed, for the 28[th] of September. The linear regression line and 'p' value are shown.

photodiode, and thus a smaller diameter to be measured. Since there was a significant step in the measured effective diameter between the 28[th] and 29[th], the most probable cause of the alignment offset was deemed to be transportation of the UCASS unit to or from the UAV launch site.

This alignment offset can be seen in Fig. 13, which is a plot of the UCASS instrument response—a 12-bit analogue to digital converter (ADC) value—against scattering cross section diameter, created from calibration data collected before and after the campaign with the same unit. A large differential can be seen between the two curves, particularly for droplets which would produce an ADC output of 1000 or higher. For example, a droplet which produces an ADC output of 3000 would be sized as $30\mu$m using the calibration data taken before the campaign, and $36\mu$m using that taken after the campaign. All the data

discussed here used the calibration conducted prior to the campaign. The re-calibration data were not applied retroactively, since there would be a combination of steady shifts and discrete steps in calibration offset which would be difficult to account for and interpolate. In addition, flights which were not analysed here were also performed after the 29[th] of September, which would have caused calibration offsets that would not be applicable to these data. The degradation in UCASS alignment over the course of the campaign points to a need for a mechanical re-design of UCASS before it's regular utilisation on fixed-wing

UAVs, similar to that conducted in Girdwood et al. (2020). However, one single unknown event—as opposed to a gradual offset over time—between the two days appears to be the source of the misalignment, since there is a large step in the effective diameter differential between the two days considered.

### 4.2.2    Measured Concentration

The concentration measured by the FMI-Talon and Static UCASS units agreed within 15% in all cases, apart from flights on

the 28[th] departing at 11:25, 12:09, and the descending profile of that departing at 12:45. The dn/dlog($D_p$) for these profiles can be seen in Fig. 10c through f, and h. In each case, the concentration measured by the FMI-Talon – UCASS was larger than that measured by the wind-vane – UCASS. This was unusual since most artefacts the FMI-Talon – UCASS could suffer



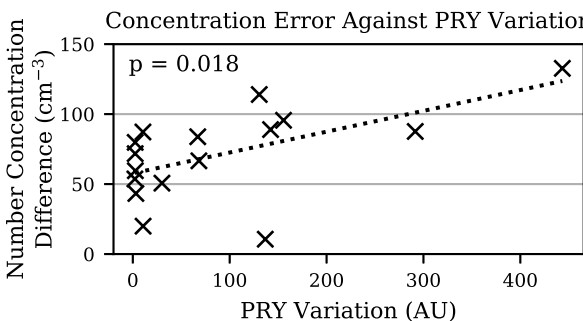

**Figure 15.** A correlation plot of the absolute difference in measured number concentration between the FMI-Talon and Static UCASS units, and PRY Variation, for the whole campaign. The linear regression line and 'p' value are shown.

from—for example large AoA values—would cause its measured concentration to decrease. Indeed regression of the absolute number concentration differential and AoA yielded 'p' values much larger than 0.05, indicating low statistical significance.

Additionally, the difference in effective diameter for the measurements conducted on the 28[th] was minimal, and AoA effects would cause a shift in the measured size distribution towards smaller sizes, as demonstrated in Sect. 3. A difference between profile type—that is, spiral versus wind-parallel lines—was also considered as a possible cause, however the flights departing at 11:26 utilised a wind-parallel profile, and the spiral descent profiles conducted on the 29[th] did not suffer from the same concentration differential.

While the regression of the absolute number concentration differential and airspeed yielded a large 'p' value over the whole campaign, when only the data for the 28[th] of September were considered, the 'p' value was 0.006—which can be seen in Fig. 14. The more rigid correlation on the 28[th] than the 29[th] indicated a dependence of the airspeed induced concentration differential on another variable, which changed between the two days. Initial observations revealed that the variation in pitch, roll, and yaw of the aircraft were much higher on the 28[th] than the 29[th]. For a quantitative definition, the root mean square

(RMS) of the signal was used. In order to remove larger oscillations in the data which occurred naturally as a response to wind speed changes, the signal was first normalised against a filtered version of itself, obtained using a moving average filter with a window size of 10 seconds. This window size was chosen because it was observed that these large variations did no occur on smaller timescales than this. This variable, $\gamma_{RMS}$, was defined mathematically as

$$\gamma_{RMS} = \sqrt{\frac{1}{T_2 - T_1} \int_{T1}^{T2} \left( \frac{f(t)}{f(t) \circledast (H(t) - H(t - w))} \right)^2} \tag{11}$$

where $T_1$ and $T_2$ are the time limits corresponding to when the UAV was at the altitude of Sammaltunturi station $\pm 20$ m; $t$ is time; $w$ is the window size in seconds, which in this instance was 20 s; $f(t)$ is the pitch, roll, and yaw (PRY) of the UAV as





a function of time; and $H(t)$ is the Heaviside function of time. This was then calculated for PRY, which were then multiplied together to give the total PRY variation.

Figure 15 shows the correlation of number concentration differential with PRY variation throughout the whole campaign. The regression yielded a 'p' value of 0.018, which indicated statistical significance. The combination of the high Reynolds number resulting from a larger airspeed, and the large PRY perturbations encountered on the 28[th], would result in an increased likelihood of turbulence. Turbulence can both cause a droplet to enter the sample area at an oblique angle, and slow down the airspeed at the sample volume. Both of these effects would cause a larger ToF and therefore concentration to be recorded. Since the pitot tube was blocked throughout the cloud there was unfortunately no other airspeed measurement to compare with, however turbulence was considered the most likely cause of the concentration differential. This result further exaggerated the need for smooth and steady sampling flight with a UAV. Since over half of the affected profiles were spiral-type, it was considered likely that the constant movement exacerbated the PRY variations. Since much of the data also had to be rejected from the spiral-type profiles, due to AoAs exceeding limitations defined in Sect. 3, the wind-parallel line profiles were considered more reliable for sampling with UCASS.

## 5 Conclusions

In this paper, a UCASS OPC was integrated on board a fixed-wing UAV—the FMI-Talon—and investigations were conducted into the accuracy and reliability of the combination. The UCASS, presented in Smith et al. (2019), utilised a gain configuration which allowed it to measure a size range of approximately $1\mu$m to $33\mu$m. Unlike in Girdwood et al. (2020), the UCASS was not re-designed for mechanical robustness. The FMI-Talon was capable of performing multiple sampling flights up to 1 km altitude, and functioning consistently throughout the day. It was also configured with temperature and humidity sensors which were not used in this research.

The investigations were composed of CFD-LPT simulations of the combinations, and a field campaign examination, involving an inter-comparison with reference instrumentation. The CFD-LPT simulations focused on two UCASS positions on the airframe. The first configuration involved the UCASS placed as far forward as the CoG tolerance of the FMI-Talon would allow. It was found that the UCASS could not be positioned forward enough to be beyond the influence of the deflection of the air by the nose cone, which caused a leading edge vortex to form on the UCASS inlet. Therefore the second configuration had to be considered, that is, positioning the UCASS far enough back so an attached airflow would be axial to UCASS. This configuration was tested for a range of AoAs and simulated droplet sizes. It was found that the original AoA limit of UCASS, $\pm 10°$, was still acceptable with the FMI-Talon in this mounting configuration. However a large uncertainty value of 45% had to be placed on the concentration of droplets measured above $30\mu$m since their larger inertia caused a deflection around the UCASS sample area. The CFD-LPT simulations also compared two different methods for calculating droplet concentration: ToF and pitot-tube. It was found that the ToF method provided a more accurate concentration, but was unreliable when used in environments with a sparse ambient number concentration.



During the field campaign, the UAV was flown—utilising a mixture of spiral and wind-parallel profiles—through a stratus
cloud up to approximately 1 km and UCASS data were collected. A static UCASS mounted on a wind vane was positioned on
the roof of a station 565 m above sea level, this provided the reference data to which the FMI-Talon – UCASS was compared.
The analysed flights were conducted on the 28th and 29th of September 2020. Data were rejected according to AoA and air-
speed criteria derived from the CFD-LPT simulations. For the measurements conducted on the 28th, the difference in effective
diameter of the size distributions between the Talon and Static UCASS units was under $2\mu$m. However, the measurements
conducted on the 29th revealed a high difference in effective diameter. This cause of this was found to be an offset in optical
alignment—which likely occurred during transportation to and from the launch site, or a hard landing—which caused less light
to reach the photodiode and therefore smaller sizes to be recorded. This was proven to be the case by re-calibrating the UCASS
unit after the campaign and observing the offset. This result pointed to a need for a mechanical re-design of UCASS, similar to
Girdwood et al. (2020), before its consistent utilisation on UAVs can be realised. Since a similar assessment of the longevity
of light-weight OPCs on UAVs has not been performed, the authors expect that this will also occur in many other instruments.

A discrepancy in number concentration between the static and UAV UCASS units was also observed on some of the profiles
conducted on the 28th, where the FMI-Talon – UCASS measured higher values than the static UCASS. The number concen-
tration discrepancy correlated well with the UAV airspeed only for the flights conducted on the 28th. The variation in pitch,
roll, and yaw given by Eq. 11 was found to correlate well with number concentration discrepancy throughout the campaign,
but only reached high values on the 28th. The concentration discrepancy was therefore determined to be caused by turbulence
induced by the combination of these two variables, which would cause droplets to enter the sample area at oblique angles which
would increase their measured ToF, and therefore concentration. This effect points to the need for steady sample flights, under
auto-pilot if possible. Auto-pilot, however, was not possible for this campaign since the pitot tube used to measure airspeed
would become blocked in the cloud.

The spiral and wind-parallel profiles were both found to be acceptable. However, more data had to be rejected from the spiral
profiles than the wind-parallel profiles due to AoA constraints, and the wind-parallel profiles appeared to induce less pitch, roll,
and yaw variations. Therefore, the wind-parallel profiles are recommended where possible.

*Author contributions.* The original draft manuscript was prepared by JG and reviewed and edited by all co-authors. The project was concep-
tualised by JG, WS, and CS. Data curation was conducted by JG and DB. Formal analysis was conducted by JG. Project supervision was the
responsibility of WS and CS. Methodology was developed by JG and DB. Validation was conducted by JG. Software was created by JG and
WS.

*Competing interests.* The authors declare that they have no conflict of interest.





*Acknowledgements.* We thank the Finnish Meteorological Institute for providing accommodation and logistical support throughout the Pallas Cloud Experiment 2020 campaign.



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
