# Peer review of "Simulation and Field Campaign Evaluation of an Optical Particle Counter on a Fixed-Wing UAV"

_Atmospheric Measurement Techniques, 2021_

## Referee Comment (RC2)

Simulation and Field Campaign Evaluation of an Optical Particle Counter on a Fixed-Wing UAV
Joseph Girdwood, Warren Stanley, Chris Stopford, and David Brus.

This is a well written and well-presented paper for which the authors should be congratulated. It has tackled head-on the difficult topic of accurate cloud droplet sampling from a moving platform.

**Section 3 - Modelling**
Modelling the flow regime around the instrument is important to understand how this will bias what is being measured - I applaud the authors for what they have done here.

This section would benefit from addressing not only the flow and pressure regimes but also what effects these may have on the temperature and humidity fields the sample experiences as it passes through the instrument. Droplets\particles\aerosol in the sample will grow and shrink so that they are in equilibrium with the surrounding continuum and to really understand the biases and uncertainties in your measurements you really need to have a handle on this. What is suggested is that the evolution of an example droplet distribution be modelled as it travels through the instrument and compared with the original.

**General Instrumentation questions**
Please could you address the effect of condensation on instrument surfaces - specifically the inner surfaces but condensation on the outer can also have an effect? For example, unless active precautions are taken then where the sample tube of an instrument sticks out into the airflow the surfaces of the sampler are cooled. In a high humidity environment, like those found in clouds, this can lead to condensation onto the chilled surfaces which can impact optical performance and the humidity field around the sampled droplets - this, in turn, can affect the droplet size distribution.

Determining the temperature and humidity of the environment being sampled and at the point of detection for the droplet\particle\aerosol is a rather important supporting measurement. Droplets\particles\aerosol will grow so that they are in equilibrium with their surroundings - equilibrium with respect to humidity at the surface of a droplet\particle\aerosol is a function of size and composition, temperature and pressure. It is best practice when stating the size of a particle (and hence the concentration) to state the RH conditions - either the measured or after growth to a reference value. It is highly recommended that this is addressed for future use.

**Section 4 - Field Campaign Testing and Validation**
Profiles were performed through a 4km thick cloud that in lines 255 and 256 is stated as "normally laterally homogeneous within the distance scales discussed in this paper". Some evidence to back up this statement is needed - some references, please.

The UAS is performing profiles and the data over these profiles compared to a ground-based measurement. There needs to be a more in-depth discussion of what is being assumed here, what variation in cloud properties may be expected in the vertical and these need to be backed up with references to the literature.

The data gathered from both the static and airborne instruments have had data processing performed to come up with concentration and effective droplet diameters. I would expect that there to be some uncertainty associated with these estimates and for this to have been discussed and the range of uncertainty shown in the figures - figures 10 and 11 in particular. In these figures, you are showing the degree to which the measurements agree so showing the uncertainty is essential.

Technical notes

Line 195:  Equation 1

      Parameter "A" in the equation is not defined in text. All other parameters in this equation
      are defined in lines 196 - 199

Figures 10 and 11

      Concentration units should be $cm^{-3}$ not $cm^3$
      dn/dlog(Dp) is used in the figure captions and also in the text but dN/dlog(Dp) is used on
      the figure axis. Need for consistency.

---

## Author Comment (AC1)

**Author Response to Reviewer #1**

*Joseph Girdwood*

We thank the reviewer for their comments and welcome their extensive experience with OPCs, and input into the manuscript.

**Response to general comments:**

We agree that more work would have to be done in order to characterise aerosol distributions with the UCASS instrument. The evaluation presented here was conducted with cloud droplets, since the underlying Mie assumptions used to compute a size are more valid, and the evaluation here is more focused on the integration of the instrument with the platform as opposed to more instrument specific parameters, like the response to various aerosols—which may be non-spherical, non-homogeneous, or be highly absorbing. As you correctly point out, this is only mentioned briefly in Sect. 4.1, therefore the manuscript has now been changed to emphasise this more strongly in the abstract. UCASS has been utilised and evaluated for the measurement of aerosol—specifically Saharan dust—in Smith *et* al. (2019) and Kezoudi *et* al. (2021), however this is ongoing research.

The aim of the CFD-LPT simulations was to define a series of UAV operating conditions, with particular focus on airspeed and AoA, in which UCASS can measure with reasonable tolerance. This has been stated in Sect. 3.1.

To provide campaign context, number concentration profiles through the cloud are presented in Fig. 8. Each profile now has an ID number which is consistent throughout the manuscript.

Since several of the authors are responsible for the initial and continued development of the UCASS instrument, future work relating to this project will be largely focused on a design overhaul of the UCASS instrument to deal with the mechanical stresses encountered during an intensive UAV field campaign. The updated UCASS will also include an extended size range, and particle-by-particle data as opposed to bins. Future work will also be conducted into evaluating the new UCASS for aerosol measurements. The conclusion has been amended to state this.

**Response to minor comments in order:**

- The sentence structure here has been revised to include revisions based on a previous comment.
- The elliptical mirror is positioned with a lens angle of 60° and has a half angle of 45°. This range of scattering angles was selected as a compromise between monotonicity of the instrument response with respect to particle optical diameter—where 90° scattering is preferable—and refractive index dependency—where forward scattering is preferable. The laser operates at 650nm, which was selected based on a size parameter for a Mie scattering regime within the size range of the instrument. Section 2.1 has been amended to reflect this.
- Figure 5 has been altered to include the aft simulation in a second panel. The text and caption have also been altered to explain this in more detail.
- The correlation coefficient is now included on both Figure 14 and 15.

**References**

*Smith, H. R., Ulanowski, Z., Kaye, P. H., Hirst, E., Stanley, W., Kaye, R., Wieser, A., Stopford, C., Kezoudi, M., Girdwood, J., Greenaway, R., and Mackenzie, R.: The Universal Cloud and Aerosol Sounding System (UCASS): a low-cost miniature optical particle counter for use in dropsonde or balloon-borne sounding systems, Atmos. Meas. Tech., 12, 6579–6599, https://doi.org/10.5194/amt-12-6579-2019, 2019.*

*Kezoudi, M., Tesche, M., Smith, H., Tsekeri, A., Baars, H., Dollner, M., Estellés, V., Bühl, J., Weinzierl, B., Ulanowski, Z., Müller, D., and Amiridis, V.: Measurement report: Balloon-borne in situ profiling of Saharan dust over Cyprus with the UCASS optical particle counter, Atmos. Chem. Phys., 21, 6781–6797, https://doi.org/10.5194/acp-21-6781-2021, 2021.*

---

## Author Comment (AC2)

**Author Response to Reviewer #2**

*Joseph Girdwood*

We thank the reviewer for their comments and welcome their input into the manuscript.

**Response to general comments:**

The concern for the diameter change of a droplet when exposed to a temperature field is completely valid. With an angle of attack of -10°, the simulated temperature of the air at the sample volume is 280.19K and the temperature of the air at the inlet was 280.23K. At the time the manuscript was written, we did not consider the 0.04K change in temperature to have a considerable effect on droplet diameter, or the relative humidity of the air. We have amended the manuscript to reflect our assumptions more clearly, since not stating them was an oversight on our behalf.

Water coating the light collection optics is an important factor to consider, not only with the UCASS instrument, but with all cloud probes. Both condensation and direct liquid deposition can attenuate the collected light, thus reducing the observed scattering cross section magnitude. Since the temperature change between the sampling volume and ambient air was 0.04K, the condensation risk here was negligible, although it was considered that, at faster airspeeds, anti-fog coatings may have needed to be applied on the collecting optics. Direct liquid deposition of water droplets onto the elliptical mirror—the largest exposed optical surface, and principal scattered light collector—was a greater concern. The UCASS elliptical mirror was designed with a surrounding circular groove in the chassis, in order to prevent the droplets—which get deposited on the inner airflow surfaces near the inlet— flowing onto the optical surfaces. The UCASS was tested with this inner chassis configuration in Smith *et* al. (2019) where it was found that droplet deposition on the optical surfaces was limited. This has been added to Sect. 2.1 for clarity.

Temperature, pressure, airspeed, aircraft GPS/attitude data, and humidity are essential parameters which accompany UCASS data. The reason why temperature and humidity data are not discussed in this paper is because lightweight sensors themselves require extensive testing and validation on UAVs, which is far beyond the scope of this paper. It is planned that the next iteration of UCASS has integrated temperature, pressure, humidity, and airspeed data, since these are essential for deriving useful data products.

Lawson *et* al. (2001) and Tsay and Jayaweera (1984) both observed Arctic stratus cloud to be laterally homogeneous, which can be assumed to be the case within the 3 km by 2 km UAV operations region discussed in this paper.

The error here is defined as one standard deviation over the vertical-spatial and temporal averaging periods for the Talon-mounted and static UCASS units respectively. This is now shown as error bars on Fig. 10 and 11.

**Response to minor comments in order:**

- A is the area around the sample area origin where a particle was considered to be sampled, 10 mm in this case. The manuscript has been amended.
- The concentration unit has been amended and dN/dlog(Dp) has been used throughout for consistency.

**References**

Smith, H. R., Ulanowski, Z., Kaye, P. H., Hirst, E., Stanley, W., Kaye, R., Wieser, A., Stopford, C., Kezoudi, M., Girdwood, J., Greenaway, R., and Mackenzie, R.: The Universal Cloud and Aerosol Sounding System (UCASS): a low-cost miniature optical particle counter for use in dropsonde or balloon-borne sounding systems, Atmos. Meas. Tech., 12, 6579–6599, https://doi.org/10.5194/amt-12-6579-2019, 2019.

Lawson, R. P., Baker, B. A., Schmitt, C. G., & Jensen, T. L. (2001). An overview of microphysical properties of Arctic clouds observed in May and July 1998 during FIRE ACE. Journal of Geophysical Research: Atmospheres, 106(D14), 14989–15014. https://doi.org/10.1029/2000JD900789.

Tsay, S.-C., & Jayaweera, K. (1984). Physical Characteristics of Arctic Stratus Clouds. Journal of Climate and Applied Meteorology, 23(4), 584–596. https://doi.org/10.1175/1520-0450(1984)023<0584:PCOASC>2.0.CO;2.